# Cross-Cultural Adaptation and Validation of the Perceptions of Empowerment in Midwifery Scale in the Spanish Context (PEMS-e)

**DOI:** 10.3390/healthcare11101464

**Published:** 2023-05-18

**Authors:** Héctor González-de la Torre, María-Isabel Hernández-Rodríguez, Alba-María Moreno-Canino, Ana-María Portela-Lomba, Miriam Berenguer-Pérez, José Verdú-Soriano

**Affiliations:** 1Research Support Unit of Insular Maternal and Child University Hospital Complex of Gran Canaria, Canary Health Service, Avda Marítima del Sur S/N, 35016 Las Palmas de Gran Canaria-Canary Islands, Spain; 2Department of Nursing, University of Las Palmas de Gran Canaria, Edificio Ciencias de la Salud, C/Blas Cabrera Felipe s/n, 35016 Las Palmas de Gran Canaria-Canary Islands, Spain; 3Department of Obstetrics and Gynaecology, Insular Maternal and Child University Hospital Complex of Gran Canaria, Canary Health Service, Avda Marítima del Sur S/N, 35016 Las Palmas de Gran Canaria-Canary Islands, Spain; iherrodj@gobiernodecanarias.org (M.-I.H.-R.); amorcan@gobiernodecanarias.org (A.-M.M.-C.);; 4Department of Community Nursing, Preventive Medicine, Public Health and History of Science, Faculty of Health Sciences, University of Alicante (UA), 03690 Alicante, Spain; miriam.berenguer@ua.es (M.B.-P.); pepe.verdu@ua.es (J.V.-S.)

**Keywords:** midwifery, surveys and questionnaires, empowerment, validation studies as topic, exploratory factor analysis

## Abstract

Midwifery empowerment is an important topic. The most widely used instrument to measure the perceived empowerment of midwives is the Perceptions of Empowerment in Midwifery Scale (PEMS), which has not been validated in Spain. The aim of this study was to translate and adapt the PEMS to the Spanish context. This research was carried out in two phases; Phase 1: Methodological study; translation, backtranslation and cross-cultural adaptation of the PEMS and pilot study on the target population (10 midwives) for evaluation of face validity. Phase 2: Cross-sectional observational study to obtain a sample for construct validation by Exploratory Factor Analysis and measurement of PEMS-e reliability. Additionally, an inferential analysis was carried out to study the possible association between several collected variables and PEMS-e subscale-scores. A total of 410 midwives from 18 Spanish regions participated in the study through an online questionnaire. An initial Spanish version of the PEMS scale was produced, demonstrating adequate face validity. A final model was produced for the PEMS-e, which included 17 items classified into two subscales (“Organizational support” and “Own skills and teamwork”) with fit indexes RMSEA = 0.062 (95%CI: 0.048–0.065) and AGFI = 0.985 (95%CI: 0.983–0.989) and Cronbach’s alpha 0.922 for the total scale. Results showed that one in four midwives had considered abandoning the profession in the last 6 months (*p* ≤ 0.001). This research suggests that Spanish midwives perceive their empowerment level as low. The PEMS-e is a valid tool with solid psychometric properties that can be used in future research to identify factors that contribute to increased empowerment among Spanish midwives and inform strategies to improve job satisfaction and retention in the profession.

## 1. Introduction

The term empowerment was initially defined by Fawcett et al. as “the process of gaining influence over events and outcomes of importance to an individual or a group” [1].

In an organizational context, two types of empowerment have been described, structural and psychological [2,3]. Power is a dynamic structure, created through formal and informal systems within the organization [4]. Laschinger et al. [5] described four sources of power within organizations: access to information, access to resources, access to opportunities and support from leaders. Structural empowerment is defined as the extent to which employees feel that they have access to such structures in the workplace [5]. This is a construct that identifies those factors that negatively influence workers’ perception of work, support and self-determination in their workplace, thus enabling their elimination [6]. Psychological empowerment refers to a subject’s motivation and feeling of competence in meeting what is expected from them at work [7]. It comprises four motivational constructs: work meaning, self-determination, competence and impact on the working environment [3,7]. It reflects the perception that a person has of their work and role within the organization [2].

Both empowerment types complement each other. The study of this topic in health organizations is important and has gained relevance in the last few years. Studies indicate that professionals’ perception of empowerment not only contributes to their satisfaction at work but also positively influences the implementation of evidence-based practice in health institutions [2,3,8].

Similarly, the interest in studying midwives’ perception of empowerment has also increased [9,10,11,12], since it has often been used as an important element to support women and their families [13]. It has been claimed that empowered, competent and trained midwives will provide quality care [14].

A relationship between midwives’ empowerment and their desire to remain in their profession has been described [15,16]. Other factors closely related to empowerment have also been identified as positively contributing to the desire for permanence, such as receiving support from superiors, having access to sufficient resources, having the ability to foster relationships with the women they care for, or perceiving self-determination and independence in their work [15,17]. Finally, a protective effect of empowerment against burnout has been described for health professionals [18,19].

All the above mentioned factors support the investigation of midwives’ perception of empowerment. The Perceptions of Empowerment in Midwifery Scale (PEMS), developed by Matthews et al. [20] in Ireland is the only tool specifically designed to measure midwives’ perception of their level of empowerment. More recently, Pallant et al. published a revised version (PEMS-R) [16]. In this version, Pallant et al. inverted the scores with respect to the original scale (higher score–higher empowerment) with the aim of a better understanding [16].

Both the PEMS and the PEMS-R have been used in several studies [10,21,22]. Validation studies of the PEMS have been published in Portuguese [23], Turkish [24] and Persian [25], while the PEMS-R has been validated in Italian [26]. Appendix A shows the characteristics of all studies, as well as the variations made on the original scale to date.

Thus, no validation study of this tool has been published in Spain and no other tool to measure Spanish midwives’ level of empowerment is available. Because of this, it has not been possible to study the level of empowerment of midwives in Spain, nor has it been possible to compare it with that reported in other countries. Therefore, the objective of this study was the translation, cross-cultural adaptation and evaluation of the psychometric properties of a Spanish version of the PEMS (PEMS-e).

## 2. Materials and Methods

This research was carried out in two phases:

Phase 1: translation and cross-cultural adaptation of the Perceptions of Empowerment in Midwifery Scale (PEMS) into Spanish, plus pilot study in the target population and calculation of face validity.

Phase 2: cross-sectional observational study to obtain a sample for the calculation of construct validation and measurement of the reliability of the first version of PEMS-e produced in phase 1. Finally, an inferential analysis was carried out to study the association between several collected variables and the scores of the final version of the PEMS-e.

### 2.1. Phase 1

#### 2.1.1. Starting Instrument

The starting tool was the original Perceptions of Empowerment in Midwifery Scale (PEMS), initially composed of 22 items, scored 1–5 on a Likert scale (1 strongly agree, 2 agree, 3 neither agree nor disagree, 4 disagree, 5 strongly disagree). In the original study, four items were removed from the scale after construct validation with factor analysis (FA), leaving the scale finally composed of 18 items, organized into three subscales called “self-determination” (6 items), “effective management” (6 items) and “practice focused on women” (6 items) [20]. However, the author suggested that the initial 22 items be included in subsequent validation studies because they corresponded to questions that midwives considered important in a previous study on the subject [9].

To calculate the scale’s score, the scores of all items in a subscale were added and the result was divided by the number of items in it (e.g., for the self-determination subscale, the sum of item scores divided by 6). In this way, comparable scores were obtained. In the original scale, the higher the score, the lower the empowerment [20]. In subsequent validations carried out by other research teams the scale was reconverted into a positive scale, with higher scores corresponding to better perception of the empowerment level [16,21,25]. In this study, the later positive scale was used for the Spanish version (PEMS-e).

#### 2.1.2. Translation and Cross-Cultural Adaptation

The translation and cross-cultural adaptation process followed the steps proposed by Sousa and Rojjanasrirat [27]. Authorization was previously obtained from the original author via email. This process was carried out in November and December 2021. Two freelance translators independently translated the English version into Spanish. One of them, a native English translator and teacher of English, had no relationship with the healthcare sector. The other was a Spanish native midwife, who had completed her studies in England and was Spanish–English bilingual since childhood. Both translators were asked to rate the difficulty of the translation with one of three options: difficult, intermediate and easy. Both translations were analyzed and discussed by the research team, to produce a unified first version (PEMS-e preliminary version 1).

Two backtranslations of the PEMS-e preliminary version 1 were assigned to two independent translators, different from the former. One of was a translation company and the other was a bilingual Spanish native translator with experience in the subject (an obstetrician). They also rated the difficulty of the translation. The research team then compared both backtranslations against the original questionnaire and discussed all discrepancies until reaching consensus. These backtranslations were also emailed to the original author (Dra. Anna Mathews) for evaluation. A PEMS-e preliminary version 2 emerged from this phase.

Finally, this later version was evaluated and compared against the original PEMS by an external researcher, who was bilingual, an expert in Research Methodology in the Health Sciences and with extensive experience in the adaptation and validation of questionnaires and measurement tools.

#### 2.1.3. Pretest and Face Validity

The PEMS-e preliminary version 2 was subjected to a pretest aimed at evaluating its viability, understandability and acceptability for the target population (face validity). The professionals who participated in this phase were chosen through non-probability convenience sampling. They had to evaluate quantitatively the pertinence-relevance of the items in the scale, by scoring each item from 1 to 4, with 1 being little pertinent-relevant and 4 very pertinent-relevant. Scores recorded for each item were then added and the result was divided by the number of participants. It was established that an average score equal to or higher than 2 indicated adequate pertinence-relevance for the item. Additionally, a qualitative evaluation was carried out where the participants were asked to assess the scale in terms of acceptability and ease of understanding and to make suggestions if they considered this appropriate. On this basis, a first version of the PEMS-e scale (PEMS-e V1) was produced.

### 2.2. Phase 2

#### 2.2.1. Design

Cross-sectional study aimed at obtaining a sample for the validation of PEMS-e and the evaluation of its psychometric properties.

#### 2.2.2. Study Population

Midwives from all over Spain, who provided direct assistance to pregnant women—either during pregnancy, childbirth or puerperium—with a minimum of 1 year working experience. Not providing direct assistance to pregnant women was considered an exclusion criterion.

#### 2.2.3. Sample Size, Sampling Procedure and Data Collection

A minimum sample size of 200 midwives was estimated as necessary to conduct an exploratory factor analysis (EFA). This calculation was based on the classic recommendation of using at least 10 subjects per item of the tool to be validated and a minimum of 200 subjects if a poly-choric data matrix is used [28,29].

Data collection took place from 7 February to 8 March 2022. An electronic form was designed according to the CROSS standards [30]. Subjects were recruited by using non-probability convenience sampling through the 18 regional midwives’ associations in Spain and the National Union of Midwives. To become a member of any of these organizations a midwife’s title is mandatory. Collaborating organizations sent personalized emails to their affiliated midwives, which included a link to the online questionnaire. In this way, only members could participate, thus ensuring the internal validity of the process. The online questionnaire was lodged on a secure platform *(Google Form^©^),* which was only active during the data collection period. The participating midwives were informed of the main objective of the study.

#### 2.2.4. Variables and Data Collection Tool

The online questionnaire was composed of two parts. The first part was created specifically to collect data on the studied variables: sex, age, marital status, way of accessing the midwifery specialty, employment status, years of professional experience as a midwife, region of work, type of center in which the main professional activity was carried out, secondary job (if any) and thoughts of leaving the profession in the last 6 months. The second part of the questionnaire was the first version of the PEMS-e scale produced in phase 1 of this research (PEMS-e V1).

#### 2.2.5. Data Analysis

A descriptive inferential analysis was carried out with the statistical software IBM© SPSS Statistics v.19.0. Qualitative variables were expressed as percentages and frequencies; quantitative variables were expressed as mean, standard deviation and minimum–maximum values.

Construct validity was evaluated with a factor analysis [29,31] using the software FACTOR Release Version 12.01.02 × 64 bits, December 2021 [32]. To estimate whether the common variance required a factor analysis, the Kaiser-Meyer-Olkin index (KMO) and the Bartlett’s statistic were used. Values over 0.75 were considered suitable for the first while *p* values ≤ 0.05 were considered statistically significant for the second. An adequacy analysis of the items was carried out according to the values of the Measure of Sampling Adequacy (MSA), where factors lower than 0.50 indicated that the corresponding item did not measure the same construct than the rest and should therefore be removed from the factorial solution [33]. The adequacy of the factorial solution was evaluated through the following indexes: Root Mean Square of Residuals (RMSR), Root Mean Square Error of Approximation (RMSEA), NonNormed Fit Index (NNFI), Comparative Fit Index (CFI), Goodness of Fit Index (GFI) and Adjusted Goodness of Fit Index (AGFI). A RMSR value of 0.05 was considered an acceptable fit; RMSEA values below 0.05 were considered a good fit and values between 0.05 and 0.08, a reasonable fit. NNFI and CFI values of 0.95 or higher and GFI and AGFI values higher than 0.90 were considered indicators of a good fit [29].

An EFA was carried out using a matrix of poly-choric correlations (according to the result of the Mardia’s test for symmetry and kurtosis); factors were extracted by unweighted least squares and oblique rotation PROMIN [28,29]. A parallel analysis was made to establish the number of factors to be retained, and the consistency of the retained factors was calculated [34]; 95% confidence intervals of the model’s measurements were calculated by using bootstrapping.

Uni-dimensionality was assessed through the indexes Unidimensional Congruence (UniCo), Explained Common Variance (ECV) and Mean of Item REsidual Absolute Loadings (MIREAL) [35]. Data can be managed as essentially unidimensional for UniCo values higher than 0.95, ECV values higher than 0.85 or MIREAL values lower than 0.30 [35].

Factor consistency was evaluated by using the ORION coefficients (Overall Reliability of fully-Informative prior Oblique N-EAP scores) [36]. Additionally, the reliability of the questionnaire was analyzed by calculating the Cronbach’s Alpha coefficient (α), the Omega coefficient (ω) and the Greatest Lower Bound (GLB) [37].

Once a final model for the final version of the PEMS-e was produced, an inferential analysis was conducted to explore the association between different variables and the resulting scores. The aim was to carry out a validation by known groups based on the study of several variables (years of professional experience, thoughts of leaving the profession in the last 6 months) as well as to evaluate the scale’s performance at a practical level. The symmetry of data distribution was assessed with the Kolmogorov-Smirnov test, where symmetric distribution of scores was verified. The parametric Student’s *T* test was used to compare means for hypothesis contrast. Associations between them were considered significant for *p*-values lower than the significance level established for this study (α = 0.05). For every studied association, the effect size was calculated according to the Hedges’ formula (Hedges g). Effect sizes between 0.2–0.5 were considered “small”, those between 0.5–0.8 were considered “moderate” and those above 0.8 were considered “large” [38].

#### 2.2.6. Ethical Aspects

This study was evaluated and approved by the Research Ethics Committee (CEI/CEIm) Hospital Dr. Negrín (Code Nº2021-492-1). At the beginning of the online questionnaire, participants were informed about its purposes and the fact that accessing its fulfillment indicated that they consented to the use of the collected data. The voluntary nature of participation and the correct use of data were guaranteed. In the data analysis, a blind matrix was used, excluding participants’ identification data.

## 3. Results

### 3.1. Phase 1

#### 3.1.1. Translation and Cross-Cultural Adaptation

The first translator considered the difficulty of the translation to be “intermediate”, while the second rated it as “easy”. The subsequent review by the research team revealed no important discrepancies or items arousing controversy. Thus, a preliminary version 1 of the PEM-e scale was produced.

The backtranslation received from the translation company was identical to the original PEMS questionnaire. In order to ensure the integrity of the validation process, the research team agreed to order a new backtranslation from a different company. This time the result was not identical to the original questionnaire and was therefore considered valid, while the former was disregarded. This new backtranslation and that made by the obstetrician were described as “easy”. A comparison between the original questionnaire and both backtranslations revealed no significant differences. The opinion of the original author was that both backtranslations were faithful and similar to the original. She did not consider it necessary to modify any item, although she showed some preference for backtranslation number 2 (made by the obstetrician). The external researcher also agreed that that version was equivalent to the original PEMS.

#### 3.1.2. Pretest

A total of 10 midwives (nine women and one man) working in the delivery room or the hospitalization area of the Insular Maternal and Child University Hospital Complex of Gran Canaria took part in the pretest. The general qualitative assessment was positive in terms of understandability and acceptability of the items, and no changes were proposed. In the quantitative assessment, scores higher than 3 were recorded for all items (Appendix A). After this process, a first version of the PEMS-e questionnaire (PEMS-e V1) was prepared.

### 3.2. Phase 2

#### 3.2.1. Sociodemographic Characteristics of the Sample

A total of 417 midwives fulfilled the online questionnaire; seven were ruled out (three because they failed to fulfill the questionnaire correctly and four because they did not meet the inclusion criteria). Thus, the final sample included 410 midwives (*n* = 410) from 18 different Spanish regions. Their mean age was 40.53 years (SD = 10. 59) (range 25–68). As for years of experience, the mean was 12.43 (SD = 9.60) (range 1–42). Results showed that 103 midwives (25.1%) had considered leaving the profession in the last 6 months. Table 1 shows the sociodemographic variables expressed as frequencies and percentages. Table 2 shows the frequencies and percentages for each of the items included in PEMS-e V1 and the score means and SDs.

#### 3.2.2. Construct Validity Analysis

Mathew et al. [20] proposed an 18-item and three subscales model of the PEMS, although they recommended that subsequent validation studies included the 22 original items described in their study. Therefore, a preliminary EFA was carried out for the total sample and for all the items without a specific number of subscales.

This EFA showed good adequacy for the data, KMO = 0.883 (95%CI: 0.743–0.862) and statistically significant Bartlett’s value (*p* ≤ 0.001). However, the fit analysis of the items suggested that five items should be removed (items 2, 3, 4, 11 and 18) with values of Measure of Sampling Adequacy (MSA) lower than 0.50 (in their confidence intervals). This finding suggested that these items did not measure the same construct as the others and supported their removal (Appendix A). In this EFA, a parallel analysis recommended a two factor-subscales solution.

A second EFA was carried out with 17 items (without the above mentioned items, which were removed due to the MSA values) and two factors-subscales. In this EFA, the KMO values and the Bartlett’s statistic indicated a better fit (KMO = 0.913 (95%CI: 0.846–0.908); Bartlett *p* ≤ 0.001), and the MSA analysis did not suggest removal of any item (values higher than 0.85 for all items) (Appendix A). The two factor-subscales solution showed 60.77% explained variance according to the parallel analysis, although this later analysis had indicated a single-factor solution. The fit values for this model were: RMSEA = 0.062 (95%CI: 0.048–0.065), NNFI = 0.980 (%CI: 0.977–0.98), CFI = 0.985 (95%CI: 0.982–0.991), GFI = 0.988 (95%CI = 0.987–0.992) and AGFI = 0.985 (95%CI: 0.983–0.989), which indicated a good fit for the model. The RMSR was 0.054 (95%CI: 0.047–0.056).

Table 3 shows the factorial loads (after rotation) of this model. All items received loads over 0.350, except item 12 (supported by colleagues), which received the lowest factorial load (0.304). According to this analysis, items 1, 5, 6, 7, 8, 9, 10 and 19 composed a first factor, which corresponded to a dimension-subscale called “Organizational Support”; while items 12, 13, 14, 15, 16, 17, 20, 21 and 22 were included in a second factor-subscale, “Own skills and teamwork”.

Additionally, and given that the parallel analysis of this EFA recommended a single-factor solution, a third EFA was carried out with 17 items and one dimension. This analysis showed that, although the KMO value was very good KMO = 0.913 (95%CI: 0.846–0.908) and the Bartlett’s statistic was significant (*p* ≤ 0.001), the fit values were poorer than in the above model: RMSEA = 0.098 (0.085–0.108), CFI = 0.956 (95%CI: 0.941–0.973) and GFI = 0.965 (95%CI = 0.956–0.977).

These results, together with the inconclusive uni-dimensionality values obtained (UniCo = 0.966 (0.937–0.985), ECV = 0.832 (0.802–0.862), MIREAL = 0.282 (0.244–0.310)), lead us to rule out a model based on a single factor-subscale.

#### 3.2.3. Reliability

ORION coefficient values were: 0.948 (95%CI: 0.931–0.958) for factor-subscale 1 and 0.912 (95%CI: 0.893–0.928) for factor-subscale 2, both showing adequate consistency, with values higher than 0.80. The total Omega coefficient was 0.922, while the GLB was 0.965; the standardized Cronbach’s alpha coefficient was 0.922 for the total scale; the Cronbach’s alpha and McDonald’s Omega coefficients of subscale 1 were 0.888 and 0.895, respectively, while those of subscale 2 were 0.843 and 0.859.

#### 3.2.4. Final Model Proposed for PEMS-e

A model composed of 17 items organized in two subscales was proposed for the final version of the PEMS-e. The subscales were named according to the items included in them and in line with the construct they measured. The first, “Organizational Support”, included 8 items related to support from supervisors/managers and access to resources and continuing education. External factors influencing midwives’ perception of empowerment should be considered. The second subscale, “Own skills and teamwork”, included nine items and concerned teamwork and recognition by other professionals, besides midwives’ own tasks. This subscale was related to a midwife’s internal factors.

The scoring system for the PEMS-e final version was independent for each subscale, similarly to the original version. The scores for each item in a subscale were added and the total was then divided by the number of items; thus, scores ranged from 1 to 5 for each subscale. The higher the score, the better the perception of the empowerment level. The final version of the PEMS-e in Spanish is available in Appendix A.

#### 3.2.5. Inferential Analysis

The mean scores of the sample of the PEMS-e final version were 3.32 (SD = 0.88) for subscale1 and 3.83 (SD = 0.63) for subscale 2. The association between the considered variables and the empowerment scores in both subscales was analyzed. Variables included in this analysis were: sex (male vs. female), access as an Internal Nursery Resident Program (Yes vs. No), thoughts of leaving the profession in the last 6 months (Yes vs. No), type of work setting (hospital vs. primary care), age (18–40 years vs. more than 40 years), years of professional experience (1 to 10 years vs. more than 10 years), secondary job (Yes vs. No) and type of training as a midwife (training in Spain vs. training abroad). Table 4 shows the means and standard deviations for each of the considered groups, the *p*-values and the effect size calculated for each association.

## 4. Discussion

This is the first validation study of a tool aimed at the evaluation of midwives’ perception of their empowerment level in Spain. This implies that empowerment has been an understudied topic in Spain.

In this study, two subscales were identified for the Spanish version of PEMS (PEMS-e), differently from validation studies in other countries, e.g., the three subscales proposed by Mathews et al. [20], the five subscales proposed for the Portuguese [23] and Persian versions [25] or the four subscales of the PEMS-R [16].

This result could be partly accounted for by differences in the approach to factor analysis. In many studies, EFAs were carried out using the popular method of factor extraction by principal component plus varimax rotation [16,20,23,25]. Currently, such combination is discouraged and this approach to FA is much criticized [28,29,39]. Moreover, in some studies, little or no information is provided on the type of EFA that has been used, which makes comparison very difficult [24,26]. In our view, there is little consistency in many of the FA conducted to date in the validation and adaptation studies of the PEMS in other countries, which raises the possibility of conducting new analyses based on current methodological recommendations for FA.

In the present study, current FA recommendations have been followed [28,29,39], including item adequacy through MSA [33,40]. It is accepted that, the larger the number of items in a certain factor, the more accurately it can be measured and the more stable the factorial solution [28,29]. Factorial solutions with factors composed of less than three items, as in the model proposed for the Portuguese validation [23], including factors with one item (Recognition within the healthcare team) or two items (Communication and professional approval), do not appear to be very reliable.

Such heterogeneity in the proposed subscale-factors is reflected in the disparity of items removed by the EFA, from the original 22 proposed by Mathews et al. [20]. In our study, items 2, 3, 4, 11 and 18 were removed from the PEMS-e final version. In a recent Persian validation, items 2, 11 and 18 were removed [25]. Notice that item 11 had already been removed by Mathews et al. [20] in their original study, although it was included in the PEMS-R [16].

The proposed model of two subscales and 17 items showed good fit and adequate reliability, with Cronbach’s alpha coefficients over 0.84 for both dimensions and over 0.90 for the total scale, i.e., higher than those reported by Mathews et al. [20].

The alpha coefficient has been widely used as the common index to evaluate the reliability of a measuring tool [37]. However, some authors suggest that other more suitable estimators should be used to evaluate internal consistency, such as the omega coefficient [37]. A further alternative index, the Greatest Lower Bound (GLB), which is not yet very widespread, is a much better indicator than the alpha coefficient [37]. Therefore, both the omega and the GLB coefficients were used to assess PEMS-e reliability in this study.

This issue was not taken into account in most PEMS validation studies up to date, except for the Persian, where omega coefficient values between 0.873 and 0.719 were calculated for every one of its five subscales (lower than those of the PEMS-e), although the value for the total scale was not provided [25]. As with the above mentioned AF, more information would be needed on the reliability of some of the PEMS validations carried out.

In one of the EFAs of this study, a parallel analysis suggested a single-factor solution. However, the adjustment values for such a model were poorer than those of the two-dimension model. Furthermore, the dimensionality analysis did not support uni-dimensionality for PEMS-e. Thus, while awaiting a confirmatory factor analysis (CFA) for our model, dimensionality can be explored from the perspective of a RASCH analysis [41,42], a methodological approach that has not been used in the PEMS and PEMS-R validation studies.

Regardless of purely methodological aspects, the construct measured by the PEMS is largely influenced by the social and historical context of midwifery in different cultures [10,43], an element that should also be considered in the comparison of results from different countries. The way to access this profession, the training program and the competences of midwives in every country are largely heterogeneous [43].

In Spain, maternity care remains largely physician-led [44,45]. Most women give birth in hospital obstetric units, where both midwives and obstetricians manage childbirth. In many centers, there is little autonomy for midwives [44,45]. This scenario may account for the fact that, in our study, midwives who studied abroad assigned lower scores to their perception of empowerment in subscale 2 (Own skills and teamwork) than midwives trained in Spain. Assumedly, they had to become adapted to the work model in our country, where midwives have low autonomy. Although some theoretical approaches exist [43], there is a lack of studies comparing aspects such as midwives’ wellbeing or their level of autonomy between countries with different contexts and cultures. The availability of a tool such as the PEMS validated in several countries can make it possible to compare the state of midwifery between countries. The Spanish validation helps in this regard.

One in four midwives participating in this study had thoughts of leaving the profession in the last 6 months. In this research, this variable was recorded in order to be validated by known groups, since the PEMS-R study by Pallant et al. [16] related the abandonment of the profession to the level of empowerment. Our findings seem to support this proposal; empowerment may be directly related to job satisfaction, which would in turn be related to the desire to remain in the profession [12,22]. This result was unexpected and worrying. The abandonment of the nursing profession, especially midwifery, is a worldwide problem [46,47,48]. The implementation of policy and strategies that increase the level of empowerment of midwives can help to alleviate this problem [49]. Further studies are needed on the issue of Spanish midwives’ intention of abandon the profession, and the PEMS-e can help here.

One of the limitations of the study is the fact that, although the sample size was larger than the required minimum (410 participants, while the required minimum was 200), it was not enough for a CFA. Notice that, although many studies do conduct EFA and CFA using the same sample (e.g., Hajiesmaello et al. [25]), this is not correct and goes against current recommendations [28,29,39]. It should also be noted that the participation of midwives from different regions was rather heterogeneous. However, although the general perception of empowerment by Spanish midwives was not equally represented across the country, the healthcare model is very similar, which supports generalization of results.

Finally, convergent validity could not be evaluated, since there is no other scale to measure midwives’ perception of empowerment validated in Spain. In the original study, Mathews et al. [20], assessed convergent validity using the Conditions of Work Effectiveness Questionnaire (CWEQ), which measures structural empowerment (though not specifically designed for midwives). Since a Spanish validation of the second version of this questionnaire (CWEQ-II) has been published [50], convergent validity of PEMS-e could be evaluated using this tool in a future research line.

## 5. Conclusions

This is the first validation study of a tool aimed at evaluation of the perception of empowerment specifically designed for midwives in Spain. The PEMS-e is composed of 17 items in two subscales and is a tool with solid psychometric properties that can be used to measure perception of empowerment by Spanish midwives. Future studies should confirm this structure with CFA. For future validations of the PEMS in other countries, the current recommendations for instrument validation should be taken into account, e.g., with regard to factor analysis.

This research suggests that Spanish midwives perceive their empowerment level as low, especially those who have had thoughts of abandoning the profession in the last 6 months. The PEMS- e is a useful tool that can help to identify which factors contribute to increasing the empowerment of midwives in Spain.

The different professional models of midwifery and the levels of training for midwives in different countries may affect their level of empowerment, which may influence key issues such as job satisfaction and retention in the profession. This is in addition to cultural differences between countries. A tool such as the PEMS validated in different settings and countries can help to compare different models of midwifery, as well as allowing for the design of policies to improve midwifery empowerment worldwide.

## Figures and Tables

**Table 1 healthcare-11-01464-t001:** Sociodemographic variables (*n* = 410).

Variable	*n* (%)
Sex	
Woman	369 (90.0%)
Man	41 (10.0%)
Marital status	
Single	178 (43.4%)
Married	202 (49.3%)
Separated-Divorced	26 (6.3%)
Widowed	4 (1.0%)
Way of accessing the midwifery specialty	
Internal Nursery Resident Program (INRP)	329 (80.2%)
Other, in Spain (before INRP)	39 (9.5%)
Other, European Union	40 (9.8%)
Other, outside the European Union	2 (0.5%)
Working status	
Employed	407 (99.3%)
Unemployed	3 (0.7%)
Type of center where the main job takes place	
Public hospital	278 (67.8%)
Private hospital	5 (1.2%)
Primary care center	119 (29.0%)
Freelancer	8 (2.0%)
Secondary job	
Yes	95 (23.2%)
No	315 (76.8%)
Thoughts of abandoning the profession in the last 6 months	
Yes	103 (25.1%)
No	307 (74.9%)

**Table 2 healthcare-11-01464-t002:** Frequencies and percentages for the item’s score included in PEMS-e V1 and the score means and Standard Deviations.

Item	Strongly Agree *n* (%)	Agree *n* (%)	Neither Agree nor Disagree *n* (%)	Disagree *n* (%)	Strongly Disagree *n* (%)	Media (Standard Deviation)
Item 1. I am valued by my manager	85 (20.7)	128 (31.2)	100 (24.4)	71 (17.3)	26 (6.3)	3.43 (1.18)
Item 2. I am an advocate for birthing women.	311 (79.9)	90 (22.0)	9 (2.2)	0 (0.0)	0 (0.0)	4.74 (0.49)
Item 3. I am involved in the midwife-led practice	273 (66.6)	100 (24.4)	27 (6.6)	8 (2.0)	2 (0.5)	4.55 (0.75)
Item 4. I do not have the skills required to carry out my role ^R^	3 (0.7)	6 (1.5)	10 (2.4)	117 (28.5)	274 (66.8)	4.59 (0.68)
Item 5. I have the back-up of my manager	75 (18.3)	129 (31.5)	109 (26.6)	71 (17.3)	26 (6.3)	3.38 (1.15)
Item 6. I am not recognized for my contribution to the care of birthing women by my manager ^R^	40 (9.8)	70 (17.1)	91 (22.2)	124 (30.2)	85 (20.7)	3.35 (1.25)
Item 7. I have adequate access to resources for birthing women in my care	60 (14.6)	182(44.4)	85 (20.7)	71 (17.3)	12 (2.9)	3.50 (1.03)
Item 8. I do not have a supportive manager ^R^	44 (10.7)	86 (21.0)	96 (23.4)	105 (25.6)	79 (19.3)	3.22 (1.27)
Item 9. I have effective communication with management	68 (16.6)	145 (35.4)	95 (23.2)	81 (19.8)	21 (5.1)	3.39 (1.13)
Item 10. I am not informed about changes in my organization that will affect my practice ^R^	49 (12.0)	92 (22.4)	105 (25.6)	108 (26.3)	56 (13.7)	3.07 (1.23)
Item 11. I am adequately educated to perform my role	222 (54.1)	157 (38.3)	21 (5.1)	8 (2.0)	2 (0.5)	4.44 (0.73)
Item 12. I have support from my colleagues	174 (42.4)	195 (47.6)	37 (9.0)	4 (1.0)	0 (0.0)	4.31 (0.68)
Item 13. I am able to say no when I judge it to be necessary	64 (15.6)	179 (43.7)	95 (23.2)	63 (15.4)	9 (2.2)	3.55 (1.00)
Item 14. I do not know what my scope of practice is ^R^	6 (1.5)	21 (5.1)	32 (7.8)	207 (50.5)	144 (35.1)	4.13 (0.87)
Item 15. I am accountable for my practice	304 (74.1)	99 (24.1)	4 (1.0)	2 (0.5)	1 (0.2)	4.71 (0.53)
Item 16. I am recognized as a professional by the medical profession	74 (18.0)	154 (37.6)	95 (23.2)	62 (15.1)	25 (6.1)	3.46 (1.13)
Item 17. I have control over my practice	110 (26.8)	217 (52.9)	42 (10.2)	34 (8.3)	7 (1.7)	3.95 (0.93)
Item 18. I empower birthing women through my practice	198 (48.3)	190 (46.3)	18 (4.4)	3 (0.7)	1 (0.2)	4.42 (0.64)
Item 19. I do not have adequate access to resources for staff education and training ^R^	33 (8.0)	85 (20.7)	92 (22.4)	153 (37.3)	47 (11.5)	3.23 (1.14)
Item 20. I have autonomy in my practice	75 (18.3)	189 (46.1)	71 (17.3)	55 (13.4)	20 (4.9)	3.60 (1.08)
Item 21. I am not listened to by members of the multidisciplinary team ^R^	19 (4.6)	70 (17.1)	119 (29.0)	150 (36.6)	52 (12.7)	3.36 (1.05)
Item 22. I am recognized for my contribution to the care of birthing women by the medical profession	53 (12.9)	164 (40.0)	110 (26.8)	60 (14.6)	23 (5.6)	3.40 (1.06)

R = Scores with inverse scores.

**Table 3 healthcare-11-01464-t003:** Rotated Loading Matrix after rotation for Model PEMS-e (17 items and two factors/subscales).

Item	Factor/Subscale 1 Organizational Support	Factor/Subscale 2 Own Skills and Teamwork
Item 1. I am valued by my manager	0.881	
Item 5. I have the back-up of my manager	0.895	
Item 6. I am not recognized for my contribution to the care of birthing women by my manager	0.828	
Item 7. I have adequate access to resources for birthing women in my care	0.370	
Item 8. I do not have a supportive manager	0.978	
Item 9. I have effective communication with management	0.870	
Item 10. I am not informed about changes in my organization that will affect my practice	0.615	
Item 12. I have support from my colleagues Widowed		0.304
Item 13. I am able to say no when I judge it to be necessary		0.796
Item 14. I do not know what my scope of practice is		0.480
Item 15. I am accountable for my practice		0.597
Item 16. I am recognized as a professional by the medical profession		0.777
Item 17. I have control over my practice		0.787
Item 19. I do not have adequate access to resources for staff education and training	0.352	
Item 20. I have autonomy in my practice		0.785
Item 21. I am not listened to by members of the multidisciplinary team		0.651
Item 22. I am recognized for my contribution to the care of birthing women by the medical profession		0.714

(Loadings lower than absolute 0.300 were omitted).

**Table 4 healthcare-11-01464-t004:** Inferential analysis for every association considered for the PEMS-e.

Variables	Factor/Subscale 1 Organizational Support	Factor/Subscale 2 Own Skills and Teamwork
	Mean (Standard Deviation)	Mean (Standard Deviation)
Sex		
Woman (*n* = 369)	3.35 (0.88)	3.84 (0.61)
Man (*n* = 41)	3.09 (0.91)	3.75 (0.75)
*p* value	0.074	0.411
Effect size (Hedges g)	0.294	0.143
Age		
18 to 40 years (n = 223)	3.15 (0.85)	3.74 (0.62)
Older than 40 years (n = 187)	3.52 (0.87)	3.93 (0.63)
*p* value	≤0.001 *	0.002 *
Effect size (Hedges g)	0.430	0.304
Years of professional experience as a midwife		
1 to 10 years (*n* = 224)	3.17 (0.84)	3.74 (0.64)
More than10 years(*n* = 186)	3.50 (0.89)	3.94 (0.60)
*p* value	≤0.001 *	0.002 *
Effect size (Hedges g)	0.382	0.321
Secondary job		
No(n = 315)	3.31 (0.90)	3.82 (0.64)
Yes(n = 95)	3.34 (0.83)	3.87 (0.59)
*p* value	0.779	0.474
Effect size (Hedges g)	0.033	0.079
Way of accessing the specialty of midwifery		
Internal Nursery Resident Program (*n* = 329)	3.30 (0.87)	3.81 (0.60)
Not Internal Nursery Resident Program (*n* = 81)	3.42 (0.93)	3.89 (0.74)
*p* value	0.242	0.401 ^a^
Effect size (Hedges g)	0.136	0.127
Training in Spain or abroad		
Training in Spain (n = 368)	3.33 (0.88)	3.86 (0.61)
Training abroad (n = 42)	3.25 (0.92)	3.59 (0.77)
p value	0.578	0.036 ^a^*
Effect size (Hedges g)	0.090	0.429
Working set		
Hospital care (n = 283)	3.25 (0.87)	3.74 (0.63)
Primary care (n = 119)	3.48 (0.90)	4.05 (0.57)
*p* value	≤0.001 *	≤0.001 *
Effect size (Hedges g)	0.261	0.505
Thought of abandoning the profession in the last 6 months		
No (*n* = 307)	3.47 (0.82)	3.96 (0.55)
Yes (*n* = 103)	2.86 (0.88)	3.45 (0.70)
*p* value	≤0.001 *	≤0.001 ^a^*
Effect size (Hedges g)	0.730	0.862

* Statistically significant *p* ≤ 0.05 (Student’s T)/a = No equal variances assumed, according to Levene’s test *p* ≤ 0.05/Hedges effect size (Hedges g); takes into account the variances and sizes of both groups; values lower than 0.2 correspond to small effects, 0.5 to intermediate effects and 0.8 to large effects.

## Data Availability

The data used in this research are confidential and are protected in a coded and anonymized database kept by the research group in accordance with Spanish regulations. However, the raw data from the PEMS-e survey (response to each item) and without the rest of the sociodemographic variables could be shared with those researchers who contact the corresponding authors if requested with a reasoned and logical request.

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
