# Peer review of "Cross-Cultural Adaptation and Validation of the Perceptions of Empowerment in Midwifery Scale in the Spanish Context (PEMS-e)"

_healthcare, 2023, doi:10.3390/healthcare11101464_

Round 1
Reviewer 1 Report
Congratulations on a very well written paper. I have included some comments in the text, but I do not think it is worth to take action on them for this particular piece of research. They may be useful for further similar research that you want to set.

The use of English is very good. I would recommend a final revision to eliminate minor mistakes (see, for instance, l. 456).
Author Response
Dear reviewer
Thank you for your time and kind comments. We appreciate your recommendations for future studies; they will be very useful to us. Evidently what happened with one of the translations was a problem that forced us to make a third translation to ensure the rigour of our research. We have learned from this and in the validation studies we are currently carrying out we are not ensuring that the translators know what we are aiming for and do not resort to old methods, as you rightly say.
The midwife translator and the obstetrician translator were not professional translators, but they were completelly bilingual since childhood (native speakers).
When we refer to the original translator we mean Dra. Anna Mathews , and we have added an explanatory note in the manuscript. Thank you very much.
Best regards
Reviewer 2 Report
First of all congratulations on your work. It was a pleasure to read. Please find the following analysis and recommendations as an opportunity to improve the manuscript's readability and interest.
Abstract
Overall the abstract is clear and capable of allowing the readers an understanding of the research question and subsequent methods used.
Please consider the following recommendations:
- The meaning of PEMS is only mentioned in lines 20-21, whilst it is first used as an abbreviation in line 19
- Mention the sample size and data collection method more clearly by stating: "A total of 410 midwives from 18 Spanish regions participated in the study through an online questionnaire."
- The sentence "A first Spanish version of the PEMS scale was produced, with an adequate facial validity" is a bit unclear. It would be better to rephrase it as: "An initial Spanish version of the PEMS scale was produced, demonstrating adequate face validity."
- If possible (due to the limitations in characters/words), include a brief statement about the implications of the study for future research and practice, such as: "The PEMS-e can be used in future research to identify factors that contribute to increased empowerment among Spanish midwives and inform strategies to improve job satisfaction and retention in the profession."
Introduction
The text effectively introduces the concept of empowerment and its two types, explains their importance in health organizations, and establishes the need to study midwives' perceptions of empowerment. It then provides information on the existing tools to measure empowerment among midwives and highlights the lack of such tools for Spanish midwives.
A few suggestions for improvement are:
- Consider dividing the first paragraph into two paragraphs to make it easier to read. The first paragraph can discuss the definition of empowerment and its two types, while the second paragraph can delve into the sources of power within organizations and how they interlink with the definitions of structural and psychological empowerment.
- In the third paragraph, clarify the relationship between empowerment and the implementation of evidence-based practice in health institutions. You could rephrase the sentence to: "Studies indicate that professionals' perception of empowerment not only contributes to their satisfaction at work but also positively influences the implementation of evidence-based practice in health institutions"
- In the paragraph discussing the PEMS and PEMS-R, consider mentioning the differences between the two scales or the reason why a revised version was created. This will provide a better context for the reader.
- When discussing validation studies for different languages, specify whether those studies used the PEMS or the PEMS-R. This will make it clearer which version has been validated in each language.
- In the final paragraph, use "thus" instead of "so far" for better flow: "Thus, no validation study of this tool has been published in Spain, and no other tool to measure Spanish midwives' level of empowerment is available."
Material and Methods
- In line 100, the number "22" should be replaced with "20" to match the three subscales' total item count of 18, as described in lines 102-103. If the original scale indeed had 22 items, you should clarify where the additional items belong.
- In lines 109-110, consider providing an example calculation to illustrate the scoring process. For example: "To calculate the scale's score, the scores of all items in a subscale were added and the result was divided by the number of items in it (e.g., for the self-determination subscale, the sum of item scores divided by 6)."
- In lines 178-180, consider mentioning the total number of regional midwives' associations and the National Union of Midwives involved in the study to give readers a sense of the scope of the recruitment effort.
- In lines 181-183, clarify whether the participating midwives were aware of the study's purpose or if they were blinded to the specific objectives of the research. This information would help readers assess the potential for response bias.
- In lines 201-202, specify why you are using model fit indices in the EFA as this is not typical and may lead to a misinterpretation of the methods used. Citing a source would also help and strengthen your option (e.g. Using Fit Statistic Differences to Determine the Optimal Number of Factors to Retain in an Exploratory Factor Analysis - 10.1177/0013164419865769).
- Although it is typical to provide some inferential results, consider explaining why you are performing this type of analysis and the reason why you chose these variables (e.g. criterion validity, external validity, identifying potential biases, establishing the practical utility of the scale).
Overall, a very good and interesting presentation of the methods used. Congratulations.
Results
No major recommendations. The text is very well written, and the tables are informative and adequate for knowledgeable readers. The authors clearly understand the statistics involved.
Discussion
The discussion segment is quite comprehensive and does a good job of contextualizing the results in the context of previous validation studies of the PEMS tool. It highlights the differences in approaches to factor analysis across various studies and the importance of adhering to current methodological recommendations. The chapter also addresses the possible reasons behind the differences in the number of subscales identified in the present study compared to other versions of the PEMS.
The authors make valid points regarding the reliability assessment using the omega coefficient and GLB, highlighting the limitations in previous validation studies. They also discuss the potential influence of cultural and social context on the construct measured by the PEMS, which is essential when comparing results across countries.
The discussion of the relationship between empowerment, job satisfaction, and thoughts of leaving the profession is well-presented, indicating that the PEMS-e could be a valuable tool for further research in this area. The authors acknowledge the limitations of their study, including the sample size and the lack of convergent validity assessment, which is a strong aspect of the chapter.
Overall, the discussion chapter is well-structured and effectively communicates the study's findings, their implications, and the limitations of the research.
Conclusion
Overall the conclusion is well-built and presents the key findings. Consider:
- Mentioning the implications of the study for practice and policy, such as informing interventions and strategies to improve midwives' empowerment, job satisfaction, and retention in the profession.
- Encouraging cross-cultural comparisons to understand how empowerment levels differ among midwives in various countries and how these differences may be related to cultural, contextual, and healthcare system factors.
- Reiterating the potential benefits of using a validated tool like the PEMS-e for studying and improving midwifery empowerment in Spain and possibly in other countries.
Author Response
First of all congratulations on your work. It was a pleasure to read. Please find the following analysis and recommendations as an opportunity to improve the manuscript's readability and interest.
Dear reviewer
Thank you very much for your time and for your comments which certainly improve this manuscript. We have tried to implement all the changes you have suggested. Thank you again and We hope you are pleased with the changes. Best regards
Abstract
Overall the abstract is clear and capable of allowing the readers an understanding of the research question and subsequent methods used.
Please consider the following recommendations:
- The meaning of PEMS is only mentioned in lines 20-21, whilst it is first used as an abbreviation in line 19
- Mention the sample size and data collection method more clearly by stating: "A total of 410 midwives from 18 Spanish regions participated in the study through an online questionnaire."
- The sentence "A first Spanish version of the PEMS scale was produced, with an adequate facial validity" is a bit unclear. It would be better to rephrase it as: "An initial Spanish version of the PEMS scale was produced, demonstrating adequate face validity."
- If possible (due to the limitations in characters/words), include a brief statement about the implications of the study for future research and practice, such as: "The PEMS-e can be used in future research to identify factors that contribute to increased empowerment among Spanish midwives and inform strategies to improve job satisfaction and retention in the profession."
Thank you very much. All your contributions have been included in the new abstract.
Introduction
The text effectively introduces the concept of empowerment and its two types, explains their importance in health organizations, and establishes the need to study midwives' perceptions of empowerment. It then provides information on the existing tools to measure empowerment among midwives and highlights the lack of such tools for Spanish midwives.
Thank you for your comments.
A few suggestions for improvement are:
- Consider dividing the first paragraph into two paragraphs to make it easier to read. The first paragraph can discuss the definition of empowerment and its two types, while the second paragraph can delve into the sources of power within organizations and how they interlink with the definitions of structural and psychological empowerment.
Done. We have divided the first paragraph as you suggested. We believe it improves the comprehension of the manuscript.
- In the third paragraph, clarify the relationship between empowerment and the implementation of evidence-based practice in health institutions. You could rephrase the sentence to: "Studies indicate that professionals' perception of empowerment not only contributes to their satisfaction at work but also positively influences the implementation of evidence-based practice in health institutions"
Done. Thank you
- In the paragraph discussing the PEMS and PEMS-R, consider mentioning the differences between the two scales or the reason why a revised version was created. This will provide a better context for the reader.
- When discussing validation studies for different languages, specify whether those studies used the PEMS or the PEMS-R. This will make it clearer which version has been validated in each language.
We have modified as suggested this part to improve the understanding of the historical evolution of this scale over time, we understand that this scale has suffered from many variations and this aspect can be difficult to understand, that is why we decided to include the table where all the studies with the PEMS-PEMS-R are included. We believe it gives additional value to this manuscript.
- In the final paragraph, use "thus" instead of "so far" for better flow: "Thus, no validation study of this tool has been published in Spain, and no other tool to measure Spanish midwives' level of empowerment is available."
Done. Thank you
Material and Methods
- In line 100, the number "22" should be replaced with "20" to match the three subscales' total item count of 18, as described in lines 102-103. If the original scale indeed had 22 items, you should clarify where the additional items belong.
This was not a mistake, but perhaps we have communicated the message wrongly. Anne Mathews initially designed a scale of 22 items, but after factor analysis she eliminated 4 items, leaving a final scale of 18 items organised in these subscales.We have completely restructured the paragraph.We hope it is better understood. Thank you for this comment.
- In lines 109-110, consider providing an example calculation to illustrate the scoring process. For example: "To calculate the scale's score, the scores of all items in a subscale were added and the result was divided by the number of items in it (e.g., for the self-determination subscale, the sum of item scores divided by 6)."
Done. We also think that this way the scoring system can be better understood.
- In lines 178-180, consider mentioning the total number of regional midwives' associations and the National Union of Midwives involved in the study to give readers a sense of the scope of the recruitment effort.
We have included the requested data. In Spain there are 18 regional associations of midwives and all of them will participate, but in one region no participant answered.
- In lines 181-183, clarify whether the participating midwives were aware of the study's purpose or if they were blinded to the specific objectives of the research. This information would help readers assess the potential for response bias.
Done. Thank you very much. You are right and this comment methodologically improves the manuscript.
- In lines 201-202, specify why you are using model fit indices in the EFA as this is not typical and may lead to a misinterpretation of the methods used. Citing a source would also help and strengthen your option (e.g. Using Fit Statistic Differences to Determine the Optimal Number of Factors to Retain in an Exploratory Factor Analysis - 10.1177/0013164419865769).
Done. We have introduced two references (thank you for your reference) .You are right that readers unfamiliar with the guidelines of modern factor analysis may be confused. It was a pleasure for us to have our manuscript reviewed by reviewers like you who have mastered factor analysis. Thank you.
-Although it is typical to provide some inferential results, consider explaining why you are performing this type of analysis and the reason why you chose these variables (e.g. criterion validity, external validity, identifying potential biases, establishing the practical utility of the scale).
We had pointed this out in the discussion (validation by known groups), but you are right that if it is stated in the method it helps the understanding, so we have implemented your proposal. Thank you
Overall, a very good and interesting presentation of the methods used. Congratulations.
Thank you very much. We very much appreciate your comment on this point, and it encourages us to try to improve our research.
Results
No major recommendations. The text is very well written, and the tables are informative and adequate for knowledgeable readers. The authors clearly understand the statistics involved.
Thank you. You are very kind.
Discussion
The discussion segment is quite comprehensive and does a good job of contextualizing the results in the context of previous validation studies of the PEMS tool. It highlights the differences in approaches to factor analysis across various studies and the importance of adhering to current methodological recommendations. The chapter also addresses the possible reasons behind the differences in the number of subscales identified in the present study compared to other versions of the PEMS.
The authors make valid points regarding the reliability assessment using the omega coefficient and GLB, highlighting the limitations in previous validation studies. They also discuss the potential influence of cultural and social context on the construct measured by the PEMS, which is essential when comparing results across countries.
The discussion of the relationship between empowerment, job satisfaction, and thoughts of leaving the profession is well-presented, indicating that the PEMS-e could be a valuable tool for further research in this area. The authors acknowledge the limitations of their study, including the sample size and the lack of convergent validity assessment, which is a strong aspect of the chapter.
Overall, the discussion chapter is well-structured and effectively communicates the study's findings, their implications, and the limitations of the research.
Thank you. Our line of research is the validation of questionnaires and we try to carry out validation processes as solid as possible, although like all research, ours has limitations that we accept and expose. In this case, the sample size greatly limited the possibility of carrying out a CFA. We will try to resolve this issue in future studies. Your kind comments help us to continue working hard in this line. Thank you again
Conclusion
Overall the conclusion is well-built and presents the key findings. Consider:
- Mentioning the implications of the study for practice and policy, such as informing interventions and strategies to improve midwives' empowerment, job satisfaction, and retention in the profession.
- Encouraging cross-cultural comparisons to understand how empowerment levels differ among midwives in various countries and how these differences may be related to cultural, contextual, and healthcare system factors.
- Reiterating the potential benefits of using a validated tool like the PEMS-e for studying and improving midwifery empowerment in Spain and possibly in other countries.
Thank you. We have tried to introduce all these points in the conclusions.
We trust that the changes implemented have improved our manuscript and will eventually allow it to be published. Thank you for your comments. Best regards
